# Effects of nutrition education and home gardening interventions on feto-maternal outcomes among pregnant women in Jimma Zone, Southwest Ethiopia: A cluster randomized controlled trial

**Melesse Niguse Kuma**[1]*, **Dessalegn Tamiru**[1], **Tefera Belachew**[1,2]

**1** Department of Nutrition and Dietetics, Jimma University, Jimma, Ethiopia, **2** School of Graduate Studies, Jimma University, Jimma, Ethiopia

* meleseniguse@gmail.com

## Abstract

### Background

Although pro-dietary practices and associated malnutrition are modifiable risk factors, they have a significant effect on maternal and neonatal health outcomes. Therefore, this study aimed to assess the effect of nutritional education and home gardening promotion on feto-maternal outcomes among pregnant women.

### Methods

A three parallel arms community-based cluster randomized controlled trial was carried out in Jimma Zone, Southwest Ethiopia from August 2020 to January 2021. Eighteen selected clusters were randomly assigned into three arms: husband (pregnant woman with her husband), peers (pregnant woman with her peers), and the controls. A total of 348 pregnant women were recruited in a 1:1:1 allocation ratio to the study arms at the baseline and 336 attended the end-line survey. Three nutrition education sessions and four varieties of vegetable seeds were provided for women in the intervention arms (husband and peers) and only routine nutrition education for the controls. Data were collected using a pretested interviewer-administered structured questionnaire. Generalized estimating equation analysis (GEE) and one-way analysis of variance (ANOVA) and Kruskal Wallis test were used to evaluate the effect of the interventions. The intervention effect estimates were obtained through a difference-in-differences approach.

### Result

In the final model, neonates born to women in the husband group were 232 g heavier than those in the control groups ($\beta = 232$, 95%CI: 228.00, 236.27. Similarly, women in the husband group had a 0.45 g/dl greater hemoglobin level than the control groups ($\beta = 0.45$, 95%

**Data Availability Statement:** All relevant data are within the manuscript and its Supporting information files.

**Funding:** The authors received no specific funding for this work.

**Competing interests:** The authors have declared that no computing interests exist.

CI: 36.48, 54.40). Likewise, a minimum diet diversity score was higher in the husband group as compared to the controls (β = 0.87 95% CI: (0.56, 1.18).

## Conclusions

Therefore, nutrition education and home gardening interventions resulted in a significant positive effect on the mean birth weight and maternal hemoglobin level among the intervention groups. The findings imply the need for enhancing such interventions to improve feto-maternal outcomes. The trial was registered at Pan African Clinical Trial Registry as PACTR202008624731801.

## Introduction

Maternal undernutrition both during early life and pregnancy is one of the proximal determinants of neonatal birth weight [1]. Evidence shows that low maternal mid-upper arm circumference (MUAC), is associated with adverse fetal outcomes [2–4]. Similarly, iron-deficiency anemia is the most common micronutrient deficiency reported during pregnancy that is known to have an impact on feto-maternal health [5, 6]. Maternal nutrition during pregnancy has an important contribution to the attainment of optimal maternal and neonatal outcomes [7].

Maternal nutrition during pregnancy has an important contribution to the attainment of optimal maternal and neonatal outcomes [8]. According to the recommendation of the World Health Organization (WHO), pregnant women should be encouraged and supported to receive adequate nutrition through the consumption of a healthy diet for optimal maternal and neonatal health outcomes [9]. Adequate maternal nutrition during pregnancy is not only vital for their health but also for the well-being of future generations [10, 11].

There are many nutritional interventions targeting pregnant women which were tested to have a significant change in improving feto-maternal outcomes [1, 8, 12, 13] Nutrition education is one of the interventions that are aimed at improving anemia, gestational weight gain (GWG), reduce the risk of low birth weight in undernourished populations through increasing daily energy, protein, and nutrient intakes [9, 12].

It was documented that nutrition education combined with home gardening programs was more effective in increasing the production and consumption of food quantity and quality, which further alleviates undernutrition and its consequences [13, 14]. In low and middle-income countries (LMICs), maternal nutrition knowledge gained during nutritional intervention alone is difficult to translate into practice due to economic, social, and cultural barriers that women encounter [15–17]. Interpersonal communication targeting women's support networks was shown to enhance their self-efficacy in achieving the recommended practice [18, 19]. To curb the barriers, engaging husbands or family members during nutrition education is more effective in the improvement of maternal knowledge and adherence to scientific advice [20–22].

In a patriarchal and agrarian society like Ethiopia, it is essential to involve husbands and other influential household members in nutrition education during pregnancy. It is also suggested that complementing nutrition education with agricultural interventions that foster an enabling environment will ensure the availability, accessibility, and utilization of diversified, safe, and nutritious food in a sustainable way [15, 23–25].

In Ethiopia, nutrition education during pregnancy is provided by health professionals at health institutions when women come for antenatal care follow-up, emphasizing the

consumption of at least one additional meal than non–pregnant states [26–28]. In addition to this, the country has developed a national nutritional policy and strategy and signed many international nutritional declarations and commitments [24, 29, 30].

However, evidence on the effect of nutritional education and home gardening interventions on feto-maternal outcomes among pregnant women in low-income countries including Ethiopia is scanty and not conclusive [31, 32]. Hence, this study aimed to assess the effect of nutrition education and home gardening promotion through the involvement of husbands and peers on feto-maternal outcomes among pregnant women in the Jimma zone, Southwest Ethiopia.

## Materials and methods

### Study design and settings

A three-arm parallel cluster-randomized, controlled, single-blinded trial was conducted to assess the effect of nutrition education and promotion of home gardening on feto-maternal outcomes in the rural district of Jimma Zone, Southwest Ethiopia, from August 2020 to January 2021. Jimma Zone is located 345 km away from the capital city of the country, Addis Ababa, in the Southwest direction. Jimma is one of the leading coffee-producing zones in Oromia Regional State, with an annual rainfall ranging between 1200–2800 mm per annum. The Zone has two well-known agro-ecological districts (mainly coffee-growing districts and food crop-growing districts). In both districts of an estimated 12135 pregnant women, about 4117 were the first trimester. A detailed explanation of the study areas has been explained previously elsewhere [33].

The sample size was estimated by G power version 3.1.9.7. Assumptions used to calculate the required sample size were: precision of 5%, power of 80%, an effect size of 0.25, and we expect a mean birth weight change of 100 gm (the primary outcome) in the intervention groups (from 2975 g to 3075 g) based on previous studies [34, 35]. Design effects of 2 and 15% non-response rates were also considered. The total sample size was 348 pregnant women (116 per each of the three study arms). The CONSORT flow chart and checklist were prepared based on consolidated standards of reporting trials [36] (Fig 1 and S1 File).

### Recruitments and randomization

Two districts (woreda) that could represent the two known agro-ecological areas of the Jimma zone were selected. Accordingly, Mana from predominantly coffee-growing districts and Seka Chekorsa from grain and crop-growing districts were selected purposively for management and logistical reasons. Then, non-adjacent clusters or kebeles (the smallest administrative units) were selected from both districts to have buffer zones. Accordingly, a total of 18 clusters (kebeles) (8 from Mana and 10 from Seka chekorsa) districts were selected. Using the randomized complete block design; the clusters (kebeles) were grouped into two blocks based on their agroecological areas (mainly coffee or grain producing). Finally, from both blocks clusters were randomly assigned to the three study arms (husband, Peers, and control). Thus, Nase, Bidaru toli, Dimtu (of Seka), Doyo toil, Gudata bula, and Lemi Lelisa clusters (kebeles) were assigned to the husband group. While, Siba bake, Buyo kechema, Meti, Bore, Bebela kosa, and Haro clusters were assigned to the peer group. Likewise, Komo hare, Gepa seden, Wakito madalu, Kamise waraba, kore Lelisa, and kenteri clusters were assigned to the control group.

Regarding the sample size allocation, it was proportionally to the population size of respective districts and kebeles (clusters). To make it more clear, of a total of 348 first-trimester pregnant women enrolled at baseline, this means 144 women were from the mana district and 204 women were from the Seka-chekorsa district. Finally, a total of 336 pregnant women (116 1st

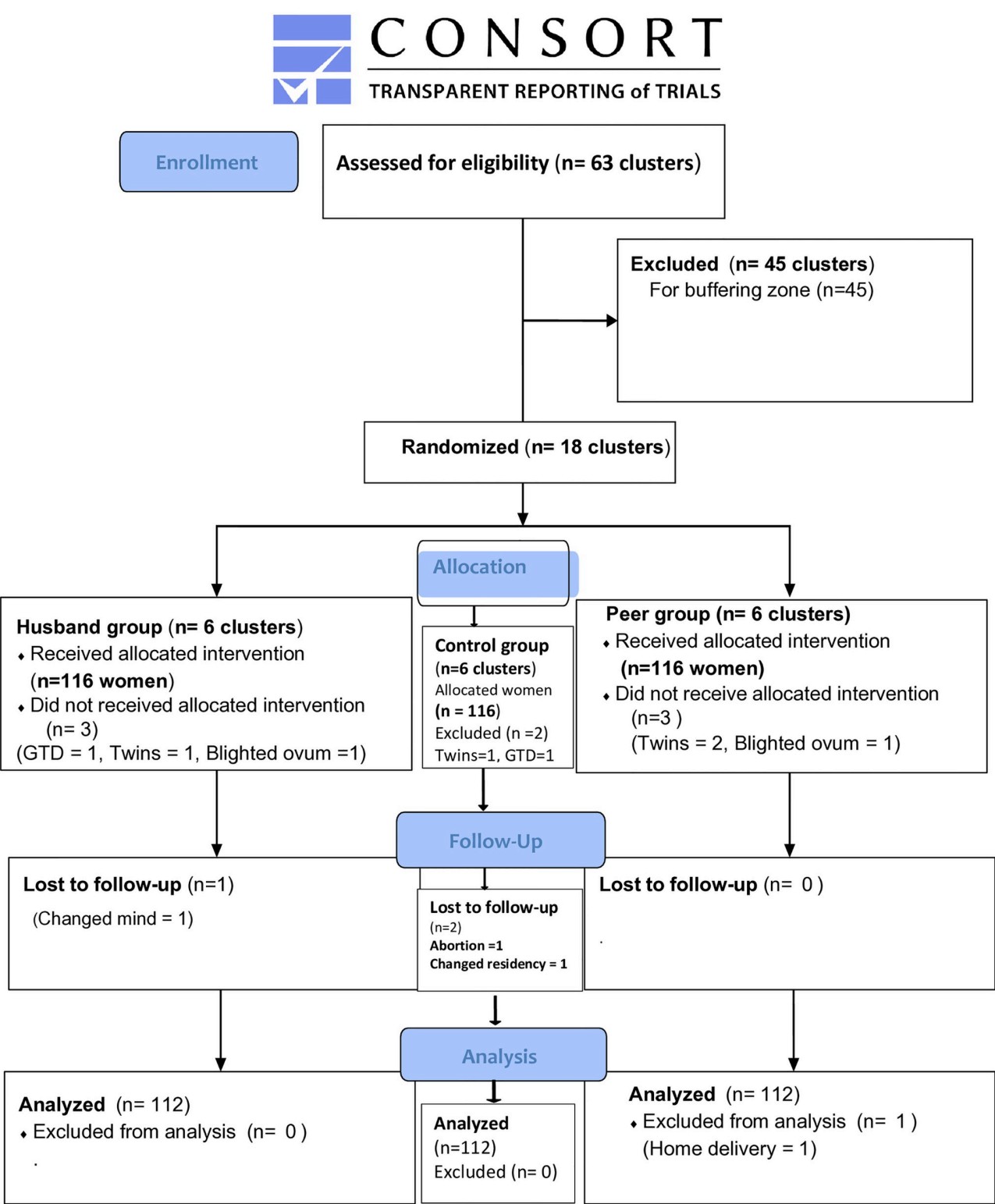

**Fig 1. The CONSORT flow diagram of the study.**

trimester pregnant women per the three study arms) were enrolled at baseline. That means an equal number of pregnant women per cluster was allocated. All of the first-trimester pregnant women who fulfilled the inclusion criteria in each cluster were enrolled until the desired or allocated number was achieved. Finally, on end-line, we have got 112 women per the study arms.

Peers are women volunteers of any age who have given birth at least once, can read and write, and are willing to complete the nutritional intervention program as well as support and mentor pregnant women in their respective clusters. Moreover, it is common in young children's feeding and prevention of mother-to-child HIV transmission (PTCT) programs [37, 38].

Eligible study participants were identified after home to a home survey of first-trimester pregnant women using the first date of their last menstrual period. Two experienced integrated emergency surgical officers (IESO) scanned the ultrasound for each enrolled woman to confirm the pregnancy and to determine the gestational age. Then, singleton first-trimester pregnant women were included in the study. Pregnancies that were identified as non-viable or had incurable deformities were excluded from the study and referred to the nearest health facilities for management.

**Blinding.** The data collectors were not informed about the allocation cluster group. Also, they were not residents of any study clusters. Due to the nature of the intervention, the study participants knew their allocation.

## Intervention

**The nutritional education intervention.** Before the onset of the intervention, training of trainers was given to eight BSc midwives for three days on the topics to be covered in each session. Maternal nutrition education intervention guideline was adapted and prepared based on the WHO recommendation of essential nutrition actions (ENA) for improving maternal nutrition guidance and maternal nutrition operational guidance for LMICs [39, 40].

The social and behavioral change communication intervention (SBC) was given to a group of pregnant women for six months in three sessions at their nearby health posts. During each SBC session, women from the husband group attended counseling with their husbands. Similarly, women from the peer group attended the intervention with their peers (the peer group was two from each cluster). Overall, the intervention was provided three times; one session per trimester. Once at the beginning of the intervention (after enrolment) in the first trimester, once during the second, and once during the third trimester.

At the end of each session, a summary of the importance of dietary diversity, especially the intake of fruits and vegetables was given as a key message. The intervention used the trans-theoretical model which stipulates individuals' motivation and readiness to change behavior over time [41]. This model assumes that people appear to go through similar stages of change no matter what behavioral therapy is being applied. But the time of change differs from individual to individual. According to the model, people progress through a series of stages of change, beginning with pre-contemplation (not intending to change) and ending with contemplation (intending to change within 6 months), followed by preparation (actively intending to change), to action (overtly making changes), and into maintenance (taking steps to sustain change) and finally termination (resisting the temptation to relapse).

In this study, the model was used to offer direction for what types of variables and processes may be important in shaping maternal nutritional health behaviors and thus need to be addressed in the intervention. The duration of education per session was 1–2 hrs. The topics of education were tested by other scholars [42]. Both the intervention and control groups

received routine nutrition education provided by antenatal care follow-up health care providers as usual.

**Topics of nutrition education delivered in different sessions.** In the first session of the intervention: the importance of dietary diversity (emphasis was given to increasing the intake of fruits and vegetables), additional meals and daytime rest, the importance of iodized salt use, the importance of weight gain, personal and environmental hygiene, iron-folic acid (IFA) supplementation and bed net use were addressed. The second session included reinforcement of the previous topics, birth preparedness, complications readiness, and danger signs of pregnancy that need immediate medical care. The final session addressed reinforcement of the previous topics, the advantage of institutional delivery, and early and exclusive breastfeeding (Table 1).

**The home gardening.** The intervention groups were supplemented with four vegetable seeds (lettuce, tomato, cabbage, and carrot), which were selected after consultation with an expert from the Jimma Zone Agricultural office. The vegetable seeds were supplied by the principal investigator. The agricultural development agents (DAs) of the clusters provided orientation training for 6 hours in the beginning and guided them from land preparation to harvesting through two weekly home visits. The purpose of providing the seeds was to motivate the participants to consume diversified home garden fruits and vegetables, not to compel the participants to eat only those vegetables (Table 1).

**Monitoring and evaluation.** Monitoring of the implementation was carried out through supervisory visits to the study area and consultative meetings with the team of investigators and the research team (counselors and peer educators). It was enhanced by recording,

**Table 1. Summary of main interventional activities for the effect of nutrition education and home gardening on feto-maternal outcomes in Jimma Zone, Southwest Ethiopia, 2020.**

| Key action(message) | | Strategy of intervention | Responsible person | Frequency | Compliance Parameter |
|---|---|---|---|---|---|
| Nutritional education | Session one | Direct information provision Sharing experiences Discussions Demonstrations Telling stories leaflets with the key message (Afan Oromo) | Trained counselors | Once during enrolment for 1-2hrs | A number of women and husbands attended nutrition education. A number of women received the leaflet. |
| | Session two | Same as above | Trained counselors | Once during the second trimester for 1–2 hours. | A number of women and husbands attended nutrition education. |
| | Session three | Same as above | Trained counselors | Once during the third trimester for 1-2hrs hours | A number of women and husbands attended the education. |
| | Home visiting | Provide information Encourage Correct | Trained counselors, peers, and PI | Monthly | The number of women visited |
| Home Gardening | Seeds provision (lettuce, tomato, cabbage, and carrot) | Provided by preparing one teaspoon of each seed/woman. | PI and DA | Once | The number of mothers who received seeds |
| | Land preparation and care for home gardens | Demonstrations | DA | Once | A number of women and their husbands attended the demonstrations |
| | Home visiting | Encourage Show Advice Correct | DA | Fortnight | The number of women prepared the land, planted the seeds, and used the vegetables |

PI; principal investigator, DA; Agricultural development agent

monitoring activities, and process evaluations. Following monitoring, necessary corrective actions were taken for contextual barriers identified during the intervention (Table 1).

**Compliance.** Mothers who attended the health education sessions and harvested home gardens were followed using a checklist. Scheduled home visits were made by the research team (counselors, DAs of their respective clusters, and peer educators).

**Outcome variables.** The neonatal birth weight and maternal hemoglobin level were the primary outcome variables of this trial, while the knowledge, attitude, and practice (KAP) scores, MDD-W score, GWG, and MUAC measurements were the secondary. The socio-economic variables and food insecurity were the independent variables.

## Data collection and measurements

The data were collected using a pretested, interviewer-administered, structured questionnaire. Data were collected by experienced and trained eight BSc holder nurses who could speak the local language, "Afan Oromo'.

**Birth weight.** The birth weight of an infant was measured within 24 hours of delivery using a digital weight scale (Seca 354) and read to the nearest 100g. Calibration of the scale to zero reading was done before each measurement and the accuracy of the scale was checked using a 1 kg weight metal [43].

**Measurement of hemoglobin.** The hemoglobin of women was measured by using Hemo-Cue AB Sweden 301(*HemoCue AB*, *Angelhom*, *Sweden*). The machine was a precalibrated instrument designed for the measurement of hemoglobin concentration. Three drops of blood were collected by a laboratory technologist from the left ring finger of each participant using a pricking method. The first and second drops of blood were wiped away and the third drop was used to test the hemoglobin level. The blood was drawn through micro cuvettes and inserted into the HemoCue machine and the results were recorded. The hemoglobin values were adjusted for altitude and trimester [44].

**Gestational weight gain (GWG).** The gestational weight was measured at the baseline and end-line. Thus, GWG was computed by subtracting the end-line from the baseline weight of the woman. The woman's weight was measured by using a digital scale (Seca 878) with a woman wearing light clothes without shoes and read to the nearest 100 gm. The scale was calibrated to zero reading before each measurement and its validity was checked using an object of 1 kg weight [45].

**Mid-upper arm circumference measurement(MUAC).** MUAC was measured using a flexible non-elastic tape to the nearest 0.1 centimeter midway between the tip of the shoulder (acromion process) and the tip of the elbow (olecranon process) of the left arm hanging freely [46]. Three measurements were taken and the average was used for final analyses.

**Minimum dietary diversity–women (MDD-W).** Data for the MDD-W score were collected by using a 24-hours dietary recall method according to the Food and Agricultural Organization's (FAO) 2016 guideline [47]. For each food group that a woman consumed, a score of "1" was given, and otherwise "0". The dietary diversity score was generated by counting the number of scores of food groups consumed. For one food group, only a score of "1" was given and summed up without considering the number of foods eaten. Finally, a woman who had gotten 5 scores out of ten or more were categorized as having adequate dietary diversity.

**Knowledge, attitude, and practice (KAP).** The woman's nutritional KAP during pregnancy was assessed by using a tool adapted from FAO 2014 guidelines for assessing nutrition-related knowledge, attitude, and practice [48]. The tool has a total of 10 items and the knowledge items address women's understanding of food groups and their sources, the

importance of a balanced diet, and the consequences of malnutrition on the fetus and the mother who is required to provide a short answer in her own words. The list of correct answers for each item includes (0) if the woman did not know any, (1) if the woman knew any one of the correct answers, and (3) if a woman knew more than one. The women's attitude was measured by asking the women to judge whether they agreed or disagree on a five-point scale. The item includes food variety, fruit, and vegetable consumption, healthy and quality foods, food taboos, rest during pregnancy, weight gain, and day rest. Moreover, the dietary practice of the women was measured using five nutrition-related practice items: changed frequency/amount, start on the additional meal, food taboo, using bed net, and daily iron folate intake. The responses were classified as (0 = no, 1 = yes). Finally, each item was summed up to generate a practice score.

**Household food insecurity access scale (HHFIAS).** Household food insecurity access was measured using nine items specific to an experience of food insecurity in the last 30-day recall period [49]. For each of the nine items, there is a follow-up question about the frequency of the occurrence (rarely, sometimes, and often). The scores ranged from 0 to 27 where the highest score reflects more food insecurity. The tool was validated for use in LMICs [50]. It was translated into the local language 'Afan Oromo'.

**Household wealth index.** It was measured using ownership of household assets: infra-structure (e.g. homes and land ownership for agricultural activities, coffee farm, electricity), small equipment (e.g. telephone, TV, radio, ornaments, plough, sofa, mattress, stove, bicycle, motorcycle), and domestic animals (cows/oxen, mule/donkey, sheep/goat). Items were coded into a relative index of household wealth using principal component analysis following a simi-lar procedure by others [51].

During the analysis the following assumptions were made; the overall sampling adequacy (KMO > 0.5), Bartlett's test of sphericity (p < 0.05), commonality > 0.5, the complex structure (correlation) not $\geq 0.40$ were checked and components that collectively explain more than 60% of the variance in the set of variables were used for generation of a continuous variable by summing up the principal components into one and tertile rank was made as rich, medium and poor.

## Data quality control

The questionnaire was translated into the local language (Afan Oromo) by language experts and then back-translated into English by another person who was blinded to the English version to check the clarity of the questionnaire. Training for three days was given to data collec-tors and their supervisors on the objective of the study, the data collection instrument, and the principles of research ethics.

A practical test on how to measure anthropometric measurement was administered to data collectors to make sure that the skill was appropriately transferred. In addition to this, two trained supervisors were assigned to give on-site support and oversee the completeness of the collected data overnight. A pretest was conducted on 5% of the total sample size in the Kersa district, a non-selected setting of the Jimma Zone. Close supervision was made by the principal investigator.

## Data processing and analysis

The data were validated, edited, coded, and entered into Epi data version 3.1 before being exported to STATA version 13 for analysis. Descriptive statistics such as frequency, percent-age, mean/ median, and standard deviation (SD) were used to describe the study subjects. The baseline difference in socio-demographic and economic characteristics between the

intervention and control arms of the study was examined using chi-square. The pre and post-intervention difference in difference (DID) estimates of maternal hemoglobin level, MUAC, and birth weight between the control and intervention groups were analyzed using one-way ANOVA after checking the assumptions. The normality assumptions and the homogeneity of variance were checked using a Q-Q plot and Levene's test. Accordingly, the maternal hemoglobin level violated the assumption of homogeneity of variance, and Welch's test was used. While, for non- normaly distributed variables: MDD-W, GWG, and maternal dietary-related KAP we used the Kruskal Wallis analyses to compare the difference in difference across the study arms. Furthermore, GEE analyses were employed to test the change difference in hemoglobin level, MUAC, MDD-W, and maternal dietary-related KAP between the interventions and control groups.

Likewise, linear GEE regression analyses were used to test the effects of the interventions on feto-maternal outcomes (birth weight and end-line hemoglobin level). The GEE was run to accommodate the correlation of observations within the subject and clustered data. The Gaussian family, identity link, and unstructured correlation matrix were considered during fitting the model and controlling for the possible potential confounding factors. Beta coefficients along with 95% confidence intervals (CI) were used to measure the effectiveness of the intervention. Interaction terms of the (intervention groups)* (time) were included to obtain the difference-in-differences effect estimates. Intention to treat (ITT) analysis was used and variables with a p-value of $< 0.05$ were considered statistically significant.

## Ethical consideration

Ethical approval was obtained from Jimma University Institutional Review Board (IRB) and Oromia Regional Health Bureau with the reference number: IHRPGD/386/19 on 29 September 2019. After explaining the objective and purpose of the study, written informed consent or thumb fingerprint (who are unable to read and write) were obtained from each participant before enrolment. Moreover, for girls (less than 18 years of age) the husband's consent with her assent was obtained. The confidentiality of the data was secured throughout the study. The privacy of the respondents was maintained by interviewing them in an isolated room. The data were kept in a locked cabinet and were not disclosed to anyone except the investigators. Before the onset of data collection, the participants were informed that the procedure constituted a minimal risk to them. The study was carried out per the Declaration of Helsinki and Good Clinical Practices [52]. The trial was registered on Pan African Clinical Trial Registry at (https://pactr.samrc.ac.za/TrialDisplay.aspx?TrialID=12278) with a PACTR202008624731801 identification number for the registry.

## Results

### Socio-demography and economic characteristics of pregnant women

A total of 348 pregnant women were recruited at the baseline from interventions (husband, n = 116, peers = 116) and control (n = 116) groups. The end-line data were collected from interventions (husband, n = 112, peers = 112) and control (n = 112) groups giving a response rate of 96.5%. The median age of the participants was 24.50, 25, and 23 years for the controls, the husband, and the peer groups, respectively. Of the total involved participants, 39(15.2%) were girls (under 18 years of age) (Table 2). Moreover, at the baseline, only half of the participants across the study arms consumed an adequate diversified diet (52.7% of controls, 50% of husbands, and 50.89% of the peer group).

**Table 2. The baseline characteristics of pregnant women in rural Southwest Ethiopia, 2020.**

| | Control | Intervention | |
| --- | --- | --- | --- |
| | (n = 112) | Husbands (n = 112) | Peers (n = 112) |
| **Number of clusters** | 6 | 6 | 6 |
| **Median age(years)** | 24.50 | 25 | 23 |
| **Mean weight(kg)** | 52.86 | 51.86 | 52.49 |
| **Gestational age(weeks)** | 11.23 | 11.60 | 11.47 |
| **Mean haemoglobin(g/dl)** | 12.55 | 12.71 | 12.52 |
| **Mean MUAC(cm)** | 23.12 | 23.05 | 23.31 |
| **Religion** | | | |
| Muslim | 99(88.39) | 103(91.96) | 100(89.28) |
| Orthodox | 10(8.92) | 6(5.35) | 12(10.71) |
| Protestant | 3(2.67) | 3(2.67) | 0(0) |
| **Maternal education** | | | |
| No formal education | 27(24.10) | 35(31.25) | 12(10.71) |
| Elementary school | 47(41.86) | 50(44.64) | 68(60.71) |
| Completed grade 8 | 15(13.39) | 13(11.60) | 15(13.39) |
| High school | 12(10.71) | 4(3.57) | 8(7.14) |
| Above high school | 11(9.82) | 10(8.92) | 9(8.03) |
| **Maternal occupation** | | | |
| Merchant | 18(16.07) | 21(18.75) | 30(26.78) |
| Housewife | 82(73.21) | 87(77.67) | 76(67.85) |
| Governmental employee | 3(2.67) | 3(2.67) | 1(0.99) |
| Student | 4(3.57) | 0(0.0) | 2(1.78) |
| Daily labourer | 5(4.46) | 1(0.89) | 3(2.67) |
| **Husband's Education** | | | |
| No formal education | 19(16.96) | 24(21.42) | 21(18.75) |
| Elementary school | 35(31.25) | 48(42.85) | 52(46.42) |
| Completed grade 8 | 17(15.17) | 14(12.50) | 17(15.17) |
| High school | 17(15.17) | 17(15.17) | 11(9.82) |
| Above high school | 24(21.42) | 9(8.03) | 11(9.82) |
| **Household head** | | | |
| Man | 106(94.64) | 110(98.21) | 110(98.21) |
| Woman | 6(5.35) | 2(1.786) | 2(1.77) |
| **Wealth index** | | | |
| Rich | 2(1.78) | 4(3.57) | 1(0.89) |
| Medium | 60(53.57) | 37(33.03) | 46(41.07) |
| Poor | 50(44.64) | 71(63.392) | 65(55.35) |
| **Districts** | | | |
| Mainly coffee-producing | 47(41.96) | 52(46.42) | 39(34.82) |
| Mainly grain-producing | 65(58.03) | 60(53.57) | 73(65.17) |

the median age value is obtained from kruskal wallis

Aberrations: MUAC; mid-upper arm circumference

## Birth weight among the study arms

The mean (± SD) birth weight of the neonates in the husband group was (3284.82 g ± 376.82), in the peer group (3181.25 g ± 401.01), and (3052.67 g ± 564.08) in the controls group. Besides, low birth weight (< 2500 g) was 5(4.46%) in the husband group, 9 (8.03%) in peers, and 27

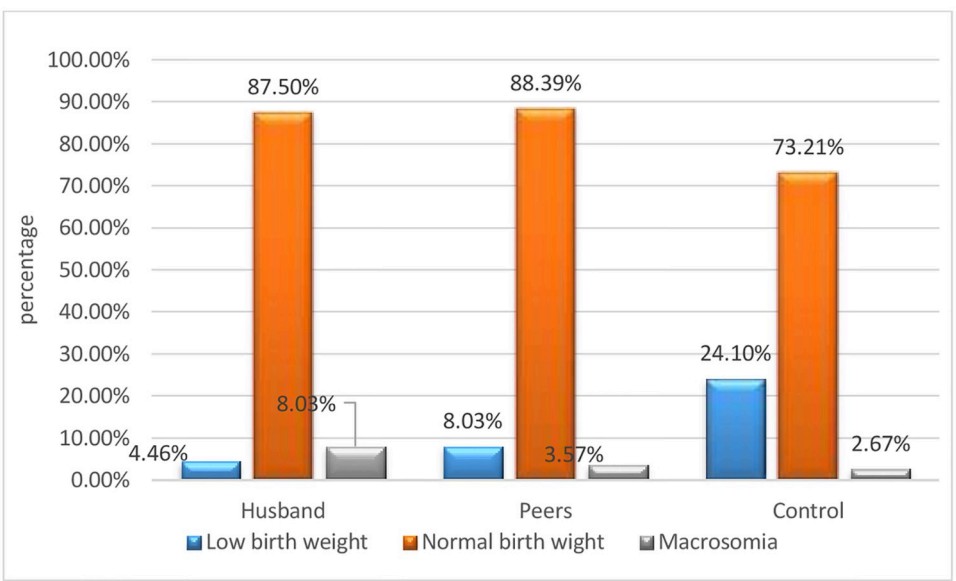

**Fig 2. Birth weight classification across the study arms among pregnant women in Jimma Zone, Southwest Ethiopia.**

(24.10%) in the control group. Moreover, macrosomia (> 4000 gm) was 9(8.03%) among the husbands' group, 4(3.57%) in the peer group, and 3(2.67%) in the control group (Fig 2).

Furthermore, the finding of ANOVA indicated that neonatal birth weight differed significantly across the arms, F (2, 333) = 7.31, P = 0.001.

Likewise, Tukey's posthoc comparisons of the three groups indicated that neonates born from the husband group (m = 3284.82 g, 95% CI: 3214.25, 3355.38) had significantly higher mean birth weight than the control group (m = 3052.67 gm, 95% CI: 2947.05, 3158.29), P = 0.001. While, comparisons between the peer group (m = 3181.25, 95% CI: 3106.17, 3256.34) and the other two groups were not significantly different. Similarly, the mean neonatal birth weight of difference-in-difference analysis revealed that neonates born from the husband group women were heavier by 232 g as compared to the neonates of the controls (232.14, 95% CI: 189.0, 275.27) p < 0.001(Table 3).

### Maternal nutrition-related knowledge, attitude, practice, and diet diversity

A comparison of the nutritional KAP scores differences among the three study arms was carried out. Hence, the Kruskal Wallis test was used to evaluate the nutritional knowledge difference of pregnant women across the study arms after the intervention. Thus, the test revealed that there is a statistically significant difference between the intervention groups and the controls *H* (2) = 106.76, *P* = < 0.001 (Table 4). Similarly, the Kruskal Wallis test showed that there is a nutritional attitude difference between the study arms *H* (2) = 24.28, *P* = < 0.001. Likewise, the test revealed that there is a statistically significant difference in nutritional practice between the pregnant women of the study arms *H* (2) = 18.26, *P* = < 0.001 (Table 4). Also, a Kruskal Wallis test showed that there is a statistically significant difference in consumption of minimum diet diversity between the study arms *H* (2) = 18.26, *P* = < 0.001 (Table 4). Also, a Kruskal Wallis test showed that there is a statistically significant difference in consumption of minimum diet diversity between the study arms *H* (2) = 18.26, *P* = < 0.001 (Table 4).

**Table 3. Comparisons of fetal birth weight and maternal hemoglobin level scores difference-in-difference across the study arms among pregnant women in Jimma Zone, Southwest Ethiopia, 2020.**

| Outcome variables | (A)Arm | (B)Arm | Mean Difference in difference(A-B) | P-value | 95% CI for mean |
|---|---|---|---|---|---|
| **Birth weight** | Husband | Peers | 103.56 | 0.21 | -39.57, 246.69 |
| | | Control | 232.14* | < 0.001 | 228.00, 236.27 |
| | Peers | Husband | -103.56 | 0.20 | -246.69, 39.57 |
| | | control | 128.58 | 0.08 | -14.55, 271.71 |
| | control | Husband | -232.14* | < 0.001 | -236.27, -228.00 |
| | | Peers | -128.58 | 0.08 | -271.71, 14.55 |
| **Hemoglobin level** | Husband | Peers | 0.05 | 0.94 | -0.35, 0.46 |
| | | Control | 0.46 | 0.02 | 0.05, 0.87 |
| | Peers | Husband | -0.05 | 0.94 | -0.46, 0.35 |
| | | Control | 0.40 | 0.05 | 0.003, 0.81 |
| | Control | Husband | -0.46 | 0.02 | -0.87, -0.05 |
| | | Peers | -0.40 | 0.05 | -0.81, 0.003 |

*. The mean difference is significant at the 0.05 level

## Comparison of nutritional status and minimum dietary diversity among groups

The ANOVA result showed that the mean hemoglobin level W (2, 219.16) = 4.69, P = 0.01 differed significantly across the study arms. Moreover, the result of Tukey posthoc comparisons indicated that the mean hemoglobin level among the women in the husbands' group (m = 0.44, 95% CI: 0.22, 0.65) was significantly higher than the control group (m = -0.02), 95% CI (-0.25, 0.20), P = 0.02 following the intervention (Table 5).

However, the mean GWG F (2,333) = 1.69, p = 0.18, and MUAC F (2,333) = 1.07, p = 0.34 were not significantly different across.

**Table 4. Kruskal Wallis Pairwise comparison of the study arms by KAP and diet diversity score.**

| Outcome Variable | Study arms | | The test statistic (H) | P- value | Adjusted p-value |
|---|---|---|---|---|---|
| **Knowledge** | Husband | Peers | 27.33 | 0.03 | 0.10 |
| | Husband | Control | 12.20 | < 0.001 | < 0.001 |
| | Peers | Control | 99.86 | < 0.001 | < 0.001 |
| **Attitude** | Husband | Peers | 8.83 | 0.49 | 1.00 |
| | Husband | Control | 58.67 | < 0.001 | < 0.001 |
| | Peers | Control | 49.83 | < 0.001 | < 0.001 |
| **Practice** | Husband | Peers | 14.36 | 0.24 | 0.73 |
| | Husband | Control | 51.13 | < 0.001 | < 0.001 |
| | Peers | control | 36.77 | 0.003 | 0.009 |
| **MDD-W** | Husband | Peers | 14.54 | < 0.001 | < 0.001 |
| | Husband | Control | 69.04 | < 0.001 | < 0.001 |
| | Peers | control | 54.50 | 0.24 | 0.74 |

Adjusted p-value; Significance values that have been adjusted by the Bonferroni correction for multiple testsAbrrvations: KAP; knowledge attitude and practice, MDD-W; minimum diet diversity of women

### Effect of the interventions on maternal hemoglobin level, MUAC, MDD-W, nutrition-related KAP

On the multivariable generalized estimating equation linear model, after controlling for possible confounding factors pregnant women of the husband group had 0.45 g/dl more hemoglobin level (β = 0.45, p = 004) as compared to the controls. Similarly, the MDD-W of pregnant women among the husband group was higher by 0.78 scores (β = 0.87, p < 0.001) as compared to the controls. Likewise, maternal nutrition-related knowledge (β = 9.75, p < 0.001), attitude (β = 1.92, p < 0.001), and practice (β = 0.69, p < 0.001) during pregnancy was significantly improved among the husband group of the intervention as compared to the controls. While, the MUAC measurements did not show a statistically significant difference between the intervention and control groups (β = 0.16, p = 0.31) (Table 5). The full set of estimates of models presented in Table 5 is attached as S3–S8 Files

## Discussion

This community-based cluster-randomized controlled trial was designed to assess the effect of nutrition education and home gardening during pregnancy on feto-maternal outcomes in Southwest Ethiopia. The findings of this study demonstrated the effectiveness of the intervention in improving the neonatal mean birth weight, maternal hemoglobin level, MDD-W, and KAP scores of the women in the husband group of the intervention arm as compared to the controls.

Thus, this trial showed that there was a significant mean birth weight difference among the study arms. Neonates born in the husband group of the study were 232 g heavier than the controls. In line with the current finding, a study from West Gojam revealed that nutritional education intervention increased the mean neonatal birth weight as compared to controls [53]. Furthermore, a study from Kenya reported that home-based monthly maternal nutrition counseling has a significantly positive effect on birth weight [54]. Similarly, a systematic review carried out on the effect of nutrition education and counseling during pregnancy indicated that it increased birth weight by 105 gm [12]. The possible explanation for this finding might be the involvement of the husband in the intervention. Because people are more likely to try a new behavior if they believe that their family, neighbors, and community will approve. Likewise, the nutritional education session given in the trimester based might increase the acceptance of the behavior to practice. These all might encourage the women to increase in the meal frequency and promotion of consumption of nutrient-rich local foods increased neonatal birth weight [55].

Similarly, after controlling the confounding factors women in the husbands' group of the intervention had significantly higher hemoglobin levels as compared to the controls. In support of this finding, a study from India showed that behavioral change communication intervention to enhance the dietary and iron-folate intake during pregnancy increased the mean hemoglobin level of the intervention group as compared to the controls [56]. Likewise, a study conducted in Timor Leste showed that husbands' knowledge of anemia is important to prevent anemia during pregnancy in their wives [57]. Furthermore, findings from Indonesia indicated that improving mothers' knowledge about pregnancy-related risks and participation of the family members, mainly husbands of less-educated mothers improved adherence to iron-folate supplementation [20]. The possible explanation is that women who attended the nutritional education with their husbands were more likely to adhere to the advice of iron-folate intake and dietary consumption which might improve their hemoglobin levels than the controls [20, 58].

**Table 5. Generalized estimating equation model predicting the effect of the intervention on hemoglobin level, MUAC, MDD-W, nutrition-related knowledge, attitude, and practice among pregnant women in Jimma Zone, Southwest Ethiopia, 2020.**

| Variables | | B | SE | P-value | 95% CI | |
|---|---|---|---|---|---|---|
| | | | | | Lower | Upper |
| **Hemoglobin level** | Intercept | 12.55 | 0.12 | < 0.001 | 12.30 | 12.80 |
| | **Groups** | | | | | |
| | Husband | 0.15 | 0.17 | 0.37 | 0.19 | 0.50 |
| | Peers | 0.03 | 0.18 | 0.85 | 0.32 | 0.38 |
| | Control | Ref | | | | |
| | Time | 0.44 | 0.10 | < 0.001 | 0.22 | 0.65 |
| | Time*husband | 0.45 | 0.15 | 0.004 | 0.36 | 0.54 |
| | Time* peer | 0.05 | 0.17 | 0.76 | 0.29 | 0.40 |
| **MUAC** | Intercept | 23.12 | 0.17 | <0.001 | 22.78 | 23.45 |
| | **Groups** | | | | | |
| | Husband | 0.006 | 0.23 | 0.77 | 0.39 | 0.52 |
| | Peer | 0.19 | 0.23 | 0.40 | 0.27 | 0.66 |
| | Control | Ref | | | | |
| | **Time** | 0.66 | 0.11 | <0.001 | 0.44 | 0.88 |
| | Time*Husband | 0.16 | 0.16 | 0.31 | 0.15 | 0.45 |
| | Time*Peer | 0.26 | 019 | 0.16 | 0.10 | 0.64 |
| **MDD-W** | Intercept | 4.53 | 0.07 | < 0.001 | 4.38 | 4.68 |
| | **Groups** | | | | | |
| | Husband | 0.02 | 0.10 | 0.80 | 0.01 | 0.2 |
| | Peer | 0.02 | 0.11 | 0.81 | 0.002 | 0.25 |
| | Control | Ref | | | | |
| | **Time** | 1.09 | 0.10 | < 0.001 | 0.88 | 1.30 |
| | Time*Husband | 0.87 | 0.15 | < 0.001 | 0.56 | 1.18 |
| | Time*Peer | 0.22 | 0.15 | 0.14 | 0.08 | 0.52 |
| **Knowledge** | Intercept | 5.09 | 0.29 | < 0.001 | 4.51 | 5.67 |
| | **Groups** | | | | | |
| | Husband | 0.009 | 0.42 | 0.98 | -0.84 | 0.83 |
| | Peers | 0.19 | 0.41 | 0.63 | -0.61 | 1.00 |
| | Control | Ref | | | | |
| | **Time** | 11.75 | 0.53 | < 0.001 | 10.71 | 12.80 |
| | Time*Husband | 9.75 | 0.81 | < 0.001 | 8.15 | 11.34 |
| | Time*Peers | 1.57 | 0.72 | < 0.03 | 0.01 | 2.99 |
| **Attitude** | Intercept | 22.72 | 0.22 | < 0.001 | 22.27 | 23.17 |
| | **Groups** | | | | | |
| | Husband | -0.17 | 0.30 | 0.57 | - 0.75 | 0.41 |
| | Peer | 0.04 | 0.31 | 0.88 | - 0.56 | 0.65 |
| | Control | Ref | | | | |
| | **Time** | 1.73 | 0.26 | < 0.001 | 1.21 | 1.25 |
| | Time*husband | 1.92 | 0.41 | < 0.001 | 1.11 | 2.74 |
| | Time*Peer | 0.16 | 0.38 | 0.67 | - 0.59 | 0.91 |

(*Continued*)

**Table 5.** (Continued)

| Variables | | B | SE | P-value | 95% CI | |
|---|---|---|---|---|---|---|
| | | | | | Lower | Upper |
| **Practice** | Intercept | 2.08 | 0.09 | < 0.001 | 1.90 | 2.27 |
| | **Groups** | | | | | |
| | Husband | 0.06 | 0.13 | 0.64 | - 0.20 | 0.32 |
| | Peer | 0.16 | 0.12 | 0.20 | - 0.08 | 0.41 |
| | Control | Ref | | | | |
| | **Time** | 2.69 | 0.10 | < 0.001 | 2.50 | 2.89 |
| | Time*husband | 0.69 | 0.16 | < 0.001 | 0.37 | 1.01 |
| | Time*Peer | 0.24 | 0.14 | 0.10 | - 0.52 | 0.04 |

the model was adjusted for maternal age, maternal education, maternal occupation, family size, household food security status, district, wealth index, alcohol consumption, and khat chewing

**Abbreviations**: β; Beta coefficients, CI; confidence interval, Ref; reference group, MDD-W; Minimum dietary diversity score of women, SE; standard error, MUAC; Mid-upper arm circumference

This study also revealed that the mean dietary diversity score of pregnant women in the husband group of the intervention arm was significantly higher compared to the control group. Similarly, another study conducted in Ethiopia also reported that pregnant women who received nutritional counseling during pregnancy were seven times more likely to consume a diversified diet than the controls [59]. Furthermore, the studies carried out in Burkina Faso and Bangladesh revealed, that there was a significant positive effect of nutrition education during pregnancy on dietary diversity scores [60, 61]. The possible explanation for the improvement in dietary diversity among the intervention groups is the detailed nature of nutritional education intervention, which encompasses the potential decision-makers of the family (the husband). Moreover, the intervention also created an enabling environment for the mothers to consume diversified food through the distribution of vegetable seeds [62].

This study also showed that maternal nutritional knowledge, attitude, and practice scores were significantly higher in the intervention groups as compared to the controls. This finding is consistent with the study carried out in Dessie Town, Northeast Ethiopia, which reported that there was a significant improvement in the mean nutritional knowledge level and dietary practices of pregnant women after attending nutritional education [63]. Similarly, a randomized controlled trial conducted in Bangladesh showed that husbands' engagement in nutrition intervention programs during their waves' pregnancies improved women's awareness, knowledge, self-efficacy, micronutrient intake, and dietary diversity [60]. Likewise, a study in Myanmar revealed that husbands who attended maternal health education with their spouses' were more involved in maternal care and support during antenatal care [21]. Similarly, a positive effect of nutritional interventions on women during pregnancy was reported in Ethiopia [59, 64], Iran [65], India [66], Indonesia [25], and Rural Bangladesh [67]. This finding might be because the method of nutritional education was simple and delivered in a culturally acceptable message were implemented in this intervention. Moreover, the involvement of the husbands and the selected peers might enable them to remember and practice it.

The findings of this study indicate that community-based nutritional education intervention during pregnancy that involves other influential community-level actors, especially husbands, combined with the promotion of locally available home garden consumption is

effective and significantly improves nutritional knowledge, attitude, and practice of pregnant women, which in turn improves maternal nutritional status and birth weight.

The strengths of this study were the strong study design used, and trimester-based community-level nutrition education, combined with agricultural interventions involving community-level actors. However, the responses to the subjective questions during data collection were dependent on maternal self-reported and memory, which might introduce some social desirability and recall biases. Though, efforts were made to probe the respondents to minimize these biases. In addition, intake was assessed using a qualitative indicator ((MDD-W score) that lacks quantification of food consumed and nutrient intake profiles.

## Conclusions

The study that, nutrition education and home gardening interventions resulted in a significantly improved mean birth weight of the infant and maternal hemoglobin level among the intervention groups. Furthermore, the intervention has a positive effect on maternal nutritional knowledge, attitude, and practice. Thus, there is a need to consider such an integrated intervention among pregnant women to improve feto-maternal outcomes.

## Supporting information

**S1 File. The CONSORT checklist of the trial.**
(DOCX)

**S2 File. Data used for the study.**
(XLSX)

**S3 File. Generalized estimating equation model predicting the effect of the intervention on hemoglobin level among pregnant women in Jimma Zone, Southwest Ethiopia, 2020.**
(DOCX)

**S4 File. Generalized estimating equation model predicting the effect of the intervention on MUAC among pregnant women in Jimma Zone, Southwest Ethiopia, 2020.**
(DOCX)

**S5 File. Generalized estimating equation model predicting the effect of the intervention on MDD-W among pregnant women in Jimma Zone, Southwest Ethiopia, 2020.**
(DOCX)

**S6 File. Generalized estimating equation model predicting the effect of the intervention on nutritional knowledge among pregnant women in Jimma Zone, Southwest Ethiopia, 2020.**
(DOCX)

**S7 File. Generalized estimating equation model predicting the effect of the intervention on nutrition related attitude among pregnant women in Jimma Zone, Southwest Ethiopia, 2020.**
(DOCX)

**S8 File. Generalized estimating equation model predicting the effect of the intervention on practice among pregnant women in Jimma Zone, Southwest Ethiopia, 2020.**
(DOCX)

**S9 File. The RCT protocol.**
(DOCX)

## Author Contributions

**Conceptualization:** Melesse Niguse Kuma, Dessalegn Tamiru, Tefera Belachew.

**Data curation:** Melesse Niguse Kuma, Tefera Belachew.

**Formal analysis:** Melesse Niguse Kuma, Dessalegn Tamiru, Tefera Belachew.

**Funding acquisition:** Melesse Niguse Kuma.

**Investigation:** Melesse Niguse Kuma, Dessalegn Tamiru, Tefera Belachew.

**Methodology:** Melesse Niguse Kuma, Dessalegn Tamiru, Tefera Belachew.

**Project administration:** Melesse Niguse Kuma, Dessalegn Tamiru, Tefera Belachew.

**Resources:** Melesse Niguse Kuma, Dessalegn Tamiru, Tefera Belachew.

**Software:** Melesse Niguse Kuma, Dessalegn Tamiru, Tefera Belachew.

**Supervision:** Melesse Niguse Kuma, Dessalegn Tamiru, Tefera Belachew.

**Validation:** Melesse Niguse Kuma, Dessalegn Tamiru, Tefera Belachew.

**Visualization:** Melesse Niguse Kuma, Dessalegn Tamiru, Tefera Belachew.

**Writing – original draft:** Melesse Niguse Kuma, Dessalegn Tamiru, Tefera Belachew.

**Writing – review & editing:** Melesse Niguse Kuma, Dessalegn Tamiru, Tefera Belachew.

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
