## [Decision Letter · Decision Letter 0]

14 Oct 2022

PONE-D-22-11496Effects of nutrition education and home gardening interventions on feto-maternal outcomes among pregnant women in Jimma Zone, Southwest Ethiopia: a cluster randomized controlled trialPLOS ONE

Dear Dr. Kuma,

Thank you for submitting your manuscript to PLOS ONE. After careful consideration, we feel that it has merit but does not fully meet PLOS ONE’s publication criteria as it currently stands. Therefore, we invite you to submit a revised version of the manuscript that addresses the points raised during the review process.

Please note that we have only been able to secure a single reviewer to assess your manuscript. We are issuing a decision on your manuscript at this point to prevent further delays in the evaluation of your manuscript. Please be aware that the editor who handles your revised manuscript might find it necessary to invite additional reviewers to assess this work once the revised manuscript is submitted. However, we will aim to proceed on the basis of this single review if possible.  Please submit your revised manuscript by Nov 28 2022 11:59PM. If you will need more time than this to complete your revisions, please reply to this message or contact the journal office at plosone@plos.org. Please include the following items when submitting your revised manuscript:A rebuttal letter that responds to each point raised by the academic editor and reviewer(s). You should upload this letter as a separate file labeled 'Response to Reviewers'.A marked-up copy of your manuscript that highlights changes made to the original version. You should upload this as a separate file labeled 'Revised Manuscript with Track Changes'.An unmarked version of your revised paper without tracked changes. You should upload this as a separate file labeled 'Manuscript'.

We look forward to receiving your revised manuscript.

Kind regards,

Avanti Dey, PhD

Staff Editor

PLOS ONE

Journal Requirements:

2. We note that in your manuscript you state that you recruited underage women (less than 18 years of age), i.e., girls, in your study. Please clarify how many girls participated in your study, and please amend your language to reflect that underage women should be referred to as 'girls'.

a) If there are ethical or legal restrictions on sharing a de-identified data set, please explain them in detail (e.g., data contain potentially identifying or sensitive patient information) and who has imposed them (e.g., an ethics committee). Please also provide contact information for a data access committee, ethics committee, or other institutional body to which data requests may be sent. Note that it is not acceptable for an author to be the sole named individual responsible for ensuring data access.

4. We note that Figure 2 in your submission contain [map/satellite] images which may be copyrighted. All PLOS content is published under the Creative Commons Attribution License (CC BY 4.0), which means that the manuscript, images, and Supporting Information files will be freely available online, and any third party is permitted to access, download, copy, distribute, and use these materials in any way, even commercially, with proper attribution. For these reasons, we cannot publish previously copyrighted maps or satellite images created using proprietary data, such as Google software (Google Maps, Street View, and Earth). For more information, see our copyright guidelines: http://journals.plos.org/plosone/s/licenses-and-copyright.

5. Please include a copy of Table 1 which you refer to in your text on page 8.

Reviewers' comments:

Reviewer's Responses to Questions

**Comments to the Author**

1. Is the manuscript technically sound, and do the data support the conclusions?

Reviewer #1: Yes

2. Has the statistical analysis been performed appropriately and rigorously? 

Reviewer #1: Yes

3. Have the authors made all data underlying the findings in their manuscript fully available?

Reviewer #1: Yes

4. Is the manuscript presented in an intelligible fashion and written in standard English?

Reviewer #1: Yes

5. Review Comments to the Author

Reviewer #1: Important note: This review pertains only to ‘statistical aspects’ of the study and so ‘clinical aspects’ [like medical importance, relevance of the study, ‘clinical significance and implication(s)’ of the whole study, etc.] are to be evaluated [should be assessed] separately/independently. Further please note that any ‘statistical review’ is generally done under the assumption that (such) study specific methodological [as well as execution] issues are perfectly taken care of by the investigator(s). This review is not an exception to that and so does not cover clinical aspects {however, seldom comments are made only if those issues are intimately / scientifically related & intermingle with ‘statistical aspects’ of the study}. Agreed that ‘statistical methods’ are used as just tools here, however, they are vital part of methodology [and so should be given due importance].

COMMENTS: Please mention all about the ‘cluster(s)’ [like cluster size, how many clusters in population, etc, etc. very briefly] in the ABTRACT [or Methods at least]. In ‘Methods’ you have just said that “A total of 348 pregnant women were recruited at the baseline and 336 attended the end-line survey” which does not define the ‘cluster(s)’. Although in lines 104-5 it is stated that “Then, non-adjacent clusters or kebeles (the smallest administrative units) were selected following the mapping of clusters by geographic information system (GIS) expertise to have buffer zones”, this may not suffice. Later in lines 109-111 you stated that “After stratifying the randomization sequence by the districts using permuted balanced block randomization (block size 18), 6 clusters were assigned to each study arm by the principal investigator” which is confusing. How do/did you stratified the randomization sequence by the districts using permuted balanced block randomization? Basically, ‘do we stratify the randomization sequence?’ is also a question.

Is/Are the cluster(s) size same? {cluster size=112/6=18.67, according to table-2 sample size of each group is 112 [or 116 according to lines 96-7: (116 per each of the three study arms)] then cluster size=116/6=19.33 & Number of clusters in each group are 6}. Is there anything wrong? How otherwise the cluster size is in fraction? You specified the block size as 18 but the number of clusters selected for each group are 6. Is not that confusing? In fact, whenever we use ‘Permuted Block Randomization’ we generally do not reveal the block size. And we generally select the block size randomly to avoid possible/easy prediction of allocation. You may know that the block size is in multiples of number of arms but it is never fixed [in advance] as we choose the block size randomly. Note that it is difficult to manage the block size as large as 18.

You may be absolutely right, in all these, however, remember that this is a scientific/academic document and so all details should be clearly/correctly communicated.

You must have tested ‘Median age(years)’ by Kruskal-Wallis’ test in Table-2. If ‘yes’, you are supposed to mention that [name of (if) another test used] in foot-note, I guess. To provide a description of baseline characteristics is entirely reasonable (since it is clearly important in assessing to whom the results of the trial can be applied), however, statistical comparison of baseline characteristics [last ‘p-value’ column in Table 2] is not desirable at all [because even if P-value turns out to be significant (while comparing baseline characteristics despite random allocation), it is, by definition, a false positive] as you then are supposed to be testing ‘randomization’ then, which in any single trial may not balance all baseline characteristics because ‘randomization’ is a sort of ‘insurance’ and not a guarantee scheme.

References:

1. Stuart J. Pocock, et al., ‘Subgroup analysis, covariate adjustment and baseline comparisons in clinical trial reporting: current practice and problems’, Statistics in medicine, 2002; 21:2917–2930 [Particularly page 2927]

2. Harrington D, et al., ‘New guidelines for statistical reporting in the journal’, N Engl J Med 2019;381:285-6

[Important message (indirectly/ultimately indicated) from these articles: Never do any comparison with respect to ‘baseline’ characteristics {by applying statistical significance test(s)}, when allocation is done randomly].

Please note that any regression techniques are not basically/originally developed for any sort of [between or within group(s)] comparison(s) including Generalized estimating equation analysis (GEE). Refer to lines 281-2: “The difference in feto-maternal outcomes (birth weight and hemoglobin level) between the intervention and control arms was examined using a linear GEE”. Otherwise using GEE is very good.

Though the measures/tools used are appropriate [examples, the Pittsburgh sleep quality index (PSQI), Household food insecurity access scale (HHFIAS), the knowledge-attitude-practice (KAP) scores, Minimum dietary diversity – women (MDD-W) score], they {few of them} may yield data that are in [at the most] ‘ordinal’ level of measurement [and not in ratio level of measurement for sure {as the score two times higher does not indicate presence of that parameter/phenomenon as double (for example, a Visual Analogue Scales VAS score or say ‘depression’ score)}]. Then application of suitable non-parametric test(s) is/are indicated/advisable [even if distribution may be ‘Gaussian’ (also called ‘normal’)]. Agreed that there is/are no non-parametric test(s)/technique(s) available to be used as alternative in all situation(s) [suitable / most desired/applicable], but should be used whenever/wherever they are available [note that generalized estimating equation analysis (GEE) is/are parametric].

I request authors to briefly explain what you mean by (line 106) “The study clusters in the area were generated using ArcGIS version 10.3” [because we generally deal with naturally formed clusters]. Although I do not completely agree with all ‘the strengths of this study’ enumerated/listed (lines 483 onwards), I definitely/confidently say that this study and the manuscript are excellent except few points highlighted above. Therefore, recommending minor revision.

6. PLOS authors have the option to publish the peer review history of their article (what does this mean?). If published, this will include your full peer review and any attached files.

Reviewer #1: No

---

## [Author Response · Author response to Decision Letter 0]

6 Feb 2023

We addressed all the comments raised by the editor as well as reviewer and wrote clarifications attached.

---

## [Decision Letter · Decision Letter 1]

21 Feb 2023

PONE-D-22-11496R1

Effects of nutrition education and home gardening interventions on feto-maternal outcomes among pregnant women in Jimma Zone, Southwest Ethiopia: a cluster randomized controlled trial

PLOS ONE

Dear Dr. Melesse Niguse Kuma

Thank you for submitting your manuscript to PLOS ONE. After careful consideration, we feel that it has merit but does not fully meet PLOS ONE’s publication criteria as it currently stands. Therefore, we invite you to submit a revised version of the manuscript that addresses the points raised during the review process.

Although I appreciate you made a good effort in addressing the reviewer's comments, I think your paper still needs clarifying and/or a different modeling approach before acceptance. Please see below:

1) You present Difference-In-Differences estimates, but the mehtods for this were not stated in the Data Analysis section. How did you get your estimates/CIs/p-values for these? These are your most important results! Which leads me to my following point...

2) Your GEE models don't seem to show DID estimates , but only main effects. I think it is crucial to get DID estimates from models, as you recognize there were very important group differences at baseline that may bias your results. The way to get DID estimates from regression models is to include an interaction term of time(baseline/endline)*treatment interaction. It seems you have not done this. I insist, this is the most important result of your work so please go through this carefully. 

Please go through the reviewer's additional comments in the attached document.

Minor issue: please use g instead of gm or gms in accordance to the International System of Units.

We look forward to receiving your revised manuscript.

Kind regards,

Hector Lamadrid-Figueroa, MD, ScD

Academic Editor

PLOS ONE

Additional Editor Comments:

Thank you for the opportunity to evaluate your excellent work. A

Reviewers' comments:

Reviewer's Responses to Questions

**Comments to the Author**

1. If the authors have adequately addressed your comments raised in a previous round of review and you feel that this manuscript is now acceptable for publication, you may indicate that here to bypass the “Comments to the Author” section, enter your conflict of interest statement in the “Confidential to Editor” section, and submit your "Accept" recommendation.

Reviewer #1: (No Response)

2. Is the manuscript technically sound, and do the data support the conclusions?

Reviewer #1: (No Response)

3. Has the statistical analysis been performed appropriately and rigorously? 

Reviewer #1: (No Response)

4. Have the authors made all data underlying the findings in their manuscript fully available?

Reviewer #1: (No Response)

5. Is the manuscript presented in an intelligible fashion and written in standard English?

Reviewer #1: (No Response)

6. Review Comments to the Author

Reviewer #1: COMMENTS: Though all the comments are answered (positively), please note that, frankly speaking, I am not very much impressed about the study & manuscript [as there are many unanswered questions regarding ‘cluster’ (size, formation/constitution, etc.), now the ABSTRACT (though is well drafted) assay type {in earlier version/draft it was in (as desired) Structured summary format and have not understood statements like (‘What exactly is the meaning of) Foot-note of table-2 which says “Footnote: the median age value is from kruskal wallis?]. Morover use of ‘English’ language is poor, making the presentation unclear/unimpressive. Therefore, I do not have any specific recommendation [though only as system requirement I choose major revision]. Let the respected editor decide the future course.

7. PLOS authors have the option to publish the peer review history of their article (what does this mean?). If published, this will include your full peer review and any attached files.

Reviewer #1: No

---

## [Decision Letter · Decision Letter 2]

12 Apr 2023

PONE-D-22-11496R2Effects of nutrition education and home gardening interventions on feto-maternal outcomes among pregnant women in Jimma Zone, Southwest Ethiopia: a cluster randomized controlled trialPLOS ONE

Dear Dr. Melesse Niguse Kuma,

Thank you for submitting your manuscript to PLOS ONE. After careful consideration, we feel that it has merit but does not fully meet PLOS ONE’s publication criteria as it currently stands. Therefore, we invite you to submit a revised version of the manuscript that addresses the points raised during the review process.

ACADEMIC EDITOR: 

Thank you for submitting your revised version. There are still some issues with your paper that need to be taken care of before considering acceptance. You included interaction terms in order to obtain actual DID estimates from your GEE models, as shown in updated table 5, but now tables 6 and 7 are redundant and make no sense, as they do not include interaction terms (DID estimates) and therefore you can not infer "the effect of the intervention" from them. Remember that all your causal claims should be based on the DID estimates. In summary:

1) please keep table 5,

2) drop tables 6 and 7 as they are redundant and

3) provide the full set of estimates of models presented in table 5 in supplementary tables.

4) Adjust your results and discussion accordingly

Please go through additional comments from reviewer 1. I will make a final evaluation of your next revision myself and reach a final decision then, so please be very thoughtful in your response/revision.

MINOR ISSUE: The phrase "non-parametric variables" makes no sense, you probably meant "non-normally distributed variables".

We look forward to receiving your revised manuscript.

Kind regards,

Hector Lamadrid-Figueroa, MD, ScD

Academic Editor

PLOS ONE

Reviewers' comments:

Reviewer's Responses to Questions

**Comments to the Author**

1. If the authors have adequately addressed your comments raised in a previous round of review and you feel that this manuscript is now acceptable for publication, you may indicate that here to bypass the “Comments to the Author” section, enter your conflict of interest statement in the “Confidential to Editor” section, and submit your "Accept" recommendation.

Reviewer #1: (No Response)

2. Is the manuscript technically sound, and do the data support the conclusions?

Reviewer #1: (No Response)

3. Has the statistical analysis been performed appropriately and rigorously? 

Reviewer #1: (No Response)

4. Have the authors made all data underlying the findings in their manuscript fully available?

Reviewer #1: (No Response)

5. Is the manuscript presented in an intelligible fashion and written in standard English?

Reviewer #1: (No Response)

6. Review Comments to the Author

Reviewer #1: COMMENTS: Though all the comments are answered {but not all attended positively} made on earlier draft, frankly speaking I am not very satisfied or convinced about few changes (with respect to many actions, very few of which are highlighted below, just for examples):

Earlier [in original manuscript] your ABSTRACT was divided in small sections like ‘Objective(s)’, ‘Methods’, ‘Results’, ‘Conclusions’, etc. which is an accepted practice of most of the good/standard journals [including this one, though ‘The PLoS One Guidelines to Authors’ did not specify an Abstract format, it is desirable]. Now the ABSTRACT {though well drafted (in my opinion), is ‘assay type’. It is preferable [refer to item 1b of CONSORT checklist 2010: Structured summary of trial design, methods, results, and conclusions] to divide the ABSTRACT. It will definitely be more informative then, I guess, whatever the article type may be.

Still the description given in ‘Recruitments and randomization’ section is confusing, particularly ‘What exactly you by “Equal allocations were given to each cluster” in line 116?’ Why used the term “Moreover,” in lines 131-32 (Moreover, due to the nature of the intervention, the study participants knew the 132 allocation)?

In lines 22-23 you stated that “Generalized estimating equation analysis (GEE) and one-way analysis of variance (ANOVA) and Kruskal Wallis test were used to evaluate the effect of the interventions” which indicates that (even the title of your study implies) that you are interested/aim at evaluating the effect of the intervention(s) but line 157 you say “This study is not designed to test any one specific theoretical model”. In my opinion this is little contradictory (according my limited knowledge of the ‘English’ language. Agreed that English is not our mother tongue (definitely not mine, may or may not be yours but certainly not of many readers), however in any case, remember/mind you that this is a scientific/academic document and so all details should be clearly/correctly communicated (do not take readers’ for granted).

Therefore, I do not have any specific recommendation [though only as system requirement I choose major revision]. Let the respected editor decide the future course. However, I request you kindly to note that I do not wish to re-review this paper/article again.

7. PLOS authors have the option to publish the peer review history of their article (what does this mean?). If published, this will include your full peer review and any attached files.

Reviewer #1: No

---

## [Author Response · Author response to Decision Letter 2]

7 May 2023

Manuscript ID: PONE-D-22-11496

Title of Manuscript: Effects of nutrition education and home gardening interventions on feto-maternal outcomes among pregnant women in Jimma Zone, Southwest Ethiopia: a cluster randomized controlled trial 

Dear Editor, 

We would like to say thank you for assigning a reviewer and returning the crucial comments. We found the points raised by the editor and reviewer very interesting to make this article scientifically sound. We addressed all the comments raised by the editor as well as reviewers and wrote clarifications here below for mutual understanding. 

Editor(s)' /Reviewers/ comments and authors' response

1. Editor(s)' Comments 

Comments and Recommendations Authors Response

1. please keep Table 5 Thank you for your comments. As per the comment given, we have kept Table 5 in the current revised manuscript 

2 Drop tables 6 and 7 as they are redundant Thank you for the concerns and important comments. We have removed Tables 6 and 7 from the revised current document.

3 Provide the full set of estimates of models presented in Table 5 in supplementary tables Thank you. We have provided the full set of models presented in Table 5 as Supplements(4-9)

4 Adjust your results and discussion accordingly Again thank you for your comments. As per the comment given we have revised the entire document and corrected the current revised manuscript.

Reviewer comments

1. Divided your manuscript into small sections like ‘Objective(s)’, ‘Methods’, ‘Results’, ‘Conclusions Thank you very much for your comments. As per the comment, a revision was made to the current document.

2. ‘What exactly do you by “Equal allocations were given to each cluster” in line 116? Thank you for your comments. Sorry for the confusion made during the edition. So, as per the comments given, we have revised it in the current document. It is to mention that each of the study arms has an equal number of pregnant women.

3. Why used the term “Moreover,” in lines 131-32? Again thank you for your comments. We have corrected in the revised current document.

4. In line 157 you say “This study is not designed to test any one specific theoretical model”.Why? Thank you very much for your valuable comments. We have corrected the current revised document as per the comment given.

Thank you in advance! 

Melesse Niguse Kuma

Corresponding author

---

## [Editor Report · Decision Letter 3]

9 May 2023

PONE-D-22-11496R3Effects of nutrition education and home gardening interventions on feto-maternal outcomes among pregnant women in Jimma Zone, Southwest Ethiopia: a cluster randomized controlled trialPLOS ONE

Dear Dr. Kuma,

Thank you for submitting your manuscript to PLOS ONE. After careful consideration, we feel that it has merit but does not fully meet PLOS ONE’s publication criteria as it currently stands. Therefore, we invite you to submit a revised version of the manuscript that addresses the points raised during the review process: Thank you for returning your revised version. I believe your paper is very close to acceptance, however you must adresss the following outstanding issues:

1. In the abstract: please briefly define the three intervention arms in the methods section.

2. In the abstract : please clarify that your effect estimates were obtained through a Difference-in-Differences approach.

3. Related to point 2: Please state in your description of the GEE statistical models in the main text, that interaction terms of the (intervention group)*(time) were included in order to obtain Difference-in-Differences effect estimates.

4. Once again: please avoid using the term "non-parametric variables" as it is innacurate. You probably mean "non-normally distributed variables".5. Please check overall grammar, punctuation, capitalization. 

We look forward to receiving your revised manuscript.

Kind regards,

Hector Lamadrid-Figueroa, MD, ScD

Academic Editor

PLOS ONE
---

## [Author Response · Author response to Decision Letter 3]

22 May 2023

Manuscript ID: PONE-D-22-11496

Title of Manuscript: Effects of nutrition education and home gardening interventions on feto-maternal outcomes among pregnant women in Jimma Zone, Southwest Ethiopia: a cluster randomized controlled trial 

Dear Editor, 

We would like to say thank you for returning the document with crucial comments. We found the points raised were very interesting to make this article scientifically sound. We addressed all the comments raised and wrote clarifications here below for mutual understanding. 

Editor(s)' /Reviewers/ comments and authors' response

1. Editor(s)' Comments 

Comments and Recommendations Authors Response

1. In the abstract: please briefly define the three intervention arms in the methods section Thank you for your comments. As per the comment given, we revised the current manuscript. # page 2 lines 19-22

2 In the abstract: please clarify that your effect estimates were obtained through a Difference-in-Differences approach Thank you very much again. We have removed in the revised current document.#page 2 lines 28 &29

3 State in your description of the GEE statistical models in the main text, that interaction terms of the (intervention group)*(time) were included in order to obtain Difference-in-Differences effect estimates. Thank you. We have included this in the revised current document.#page 18 lines 306 &307

4 Once again: please avoid using the term "non-parametric variables" as it is inaccurate. Again thank you for your comments. As per the comment given we have revised the current manuscript. # page17 line 295 & 296.

5 Please check overall grammar, punctuation, and capitalization. Thank you very much for your comments. As per the comment given revision was made to the current document and corrected as possible.

Thank you in advance! 

Melesse Niguse Kuma

Corresponding author

---

## [Editor Report · Decision Letter 4]

21 Jun 2023

Effects of nutrition education and home gardening interventions on feto-maternal outcomes among pregnant women in Jimma Zone, Southwest Ethiopia: a cluster randomized controlled trial

PONE-D-22-11496R4

Dear Dr. Melesse Niguse Kuma

We’re pleased to inform you that your manuscript has been judged scientifically suitable for publication and will be formally accepted for publication once it meets all outstanding technical requirements.

Kind regards,

Hector Lamadrid-Figueroa, MD, ScD

Academic Editor

PLOS ONE

Additional Editor Comments (optional):

Please make sure to substitute the term "Non-randomly distributed" to "Non-normally distributed". Congratulations!
---

## [Editor Report · Acceptance letter]

29 Jun 2023

PONE-D-22-11496R4 

Effects of nutrition education and home gardening interventions on feto-maternal outcomes among pregnant women in Jimma Zone, Southwest Ethiopia: a cluster randomized controlled trial 

Dear Dr. Kuma:

I'm pleased to inform you that your manuscript has been deemed suitable for publication in PLOS ONE. Congratulations! Your manuscript is now with our production department. 

Kind regards, 

on behalf of

Dr. Hector Lamadrid-Figueroa 

Academic Editor

PLOS ONE